# Semi-supervised learning in perivascular space segmentation using MRI images

**Yaqiong Chai**[1]                                      YAQIONGC@USC.EDU
**Hedong Zhang**[1*]                                     HEDONGZH@USC.EDU
**Gilsoon Park**[1]                                      GP_446@USC.EDU
**Erika Lopez**[1,3]                                     ERIKALOPEZ2800@GMAIL.COM
**Cong Zang**[1,2]                                       CONGZANG@USC.EDU
**Jongmok Ha**[1,4]                                      JONGMOK3245@GMAIL.COM
**Omar Elhawary**[1,3]                                   ELHAWARY@USC.EDU
**Hosung Kim**[1]                                        HOSUNGKI@USC.EDU

[1] *Mark and Mary Stevens Neuroimaging and Informatics Institute, Keck School of Medicine, University of Southern California (USC), Los Angeles, CA, U.S.*

[2] *Neuroscience Graduate Program, USC, Los Angeles, CA, U.S.*

[3] *Dornsife College of Letters, Art and Sciences, USC, Los Angeles, CA, U.S.*

[4] *Department of Neurology, Samsung Medical Center, Seoul, South Korea*

[*] *Corresponding author*

## Abstract

Accurate segmentation of perivascular space (PVS) is essential for its quantitative analysis and clinical applications. Various segmentation methods have been proposed, but semi-supervised learning methods have never been attempted. Here, we propose a 3D multi-channel, multi-scale semi-supervised PVS segmentation (M2SS-PVS) network. We incorporated multi-scale image features in the encoder and applied a few strategies to mitigate class imbalance issue. Our M2SS-PVS network segmented PVS with the highest accuracy and high sensitivity among all the tested supervised and semi-supervised methods.

**Keywords:** semi-supervised learning, perivascular space; MRI.

## 1. Introduction

Perivascular spaces (PVSs) are extracellular spaces containing interstitial or cerebrospinal fluid surrounding cerebral small penetrating arteries and veins (Jessen et al., 2015). PVS is visible on brain MRI scans of healthy individuals but also considered a hallmark of aging and an early anomaly of neuro-degenerative disease as it becomes enlarged along with aging and Parkinson's disease, multiple sclerosis, and small vessel disease (Zhu et al., 2010).

PVS burden is generally evaluated using semi-quantitative rating scales on T1- or T2-weighted MRI (Potter et al., 2015). Manual detection and delineation of PVS are currently the gold standard for neuro-radiological evaluation, but it is highly labor-intensive and subject to inter-rater bias. Early automated approaches were based on unsupervised methods including simple thresholding, edge detection, or morphological operations but yielded poor results (Uchiyama et al., 2008). More recent studies have adopted supervised methods for PVS segmentation. For example, Lan H. et al. incorporated a convolutional neural network (CNN) into weakly supervised learning framework and achieved a $F_\beta$ score of 0.76

within the region of interests, requiring a further improvement (Lan et al., 2023). However, PVS segmentation is very challenging due to 1) various shape and size of PVS; 2) class-imbalance issues. In addition, supervised deep learning generally requires a large size of training samples with labels.

Therefore, we propose a 3D multi-channel (T1, T2 MRI), multi-scale semi-supervised perivascular space segmentation (M2SS-PVS) network, to leverage the information contained in a larger number of unlabeled images. To evaluate the performance of our proposed M2SS-PVS network, we computed three metrics: DSC, the absolute percentage volume difference (AVD), and recall among M2SS-PVS, supervised multi-scale and mono-scale feature extraction, and semi-supervised learning with mono-scale feature extraction.

## 2. Materials and Method

**Materials.** In this work, we used T1 and T2 turbo spin echo images (both in the resolution of 0.8 $mm^3$) in the Human Connectome Project in Aging (HCP-A) (Bookheimer et al., 2019). Our subjects were sampled considering the age range of 45 years or older (63.8±14.1) and sex balance (18F). Twelve out of 33 subjects were manually segmented for PVS by two medical students reviewing both T1 and T2 images and evaluated by a neurologist.

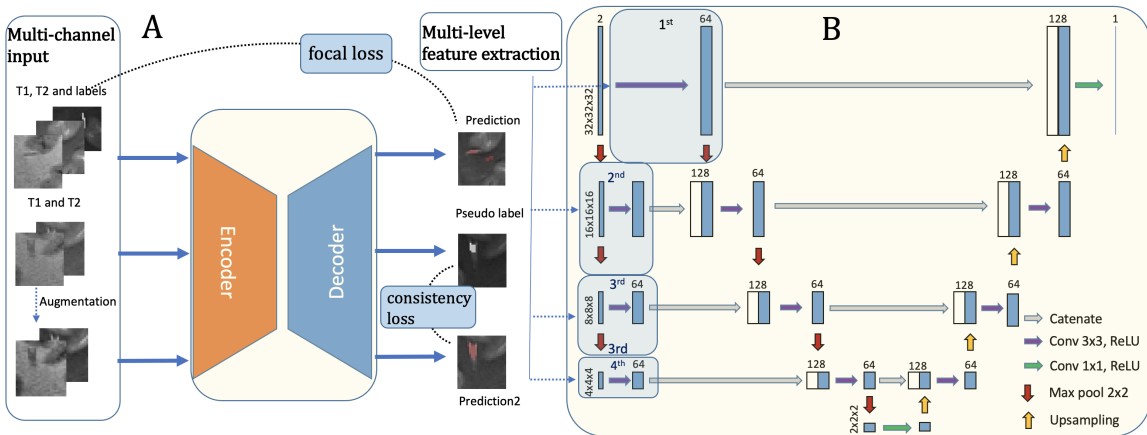

Figure 1: The architecture of the proposed M2SS-PVS network. It consists of an encoder and decoder network. A: The network follows an encoder-decoder structure. The supervised learning part is optimized by focal loss. In the unsupervised part, the pseudo label produced by T1 and T2 and the predicted results produced by the augmented images are optimized by consistency loss. B: We designed multi-scale feature extraction in the encoder to incorporate different level of features in the successive layers.

**Network Architecture.** The framework of the proposed M2SS-PVS network is illustrated in Fig.1. The supervised model is fed with labeled T1 and T2 images in the two complementary channels, and the unsupervised model is fed with unlabeled T1 and T2 images as well as those pairs augmented by applying standard Gaussian noise. The supervised loss was designed as: $\mathcal{L}_s = \frac{1}{N} \sum_{i=1}^{N} \textbf{focal loss}(P_{label}, P_{pred})$, where $P_{pred} = \mathcal{M}odel(x_{label})$. Focal loss (Lin et al., 2017) is designed to mitigate imbalanced class learning and optimize the supervised model. A consistency loss (binary cross entropy) was calculated between the pseudo labels (on the original unlabeled images) and the PVS predicted on the augmented im-

ages to optimize the unsupervised model: $\mathcal{L}_u = \frac{1}{N_u} \sum_{i=1}^{N_u} \mathbb{1}[P'_{pred} > \sigma] \mathbf{BCE}(P'_{pseudo}, P'_{pred})$, where $P'_{pseudo} = \mathcal{M}odel(x_{unlabel})$, $P'_{pred} = \mathcal{M}odel(\mathcal{A}ugmentation(x_{unlabel}))$, $\mathbb{1}$ is the indicator function and $\sigma$ is the confidence threshold ($\sigma = 0.8$) to filter pseudo labels.

To robustly capture the various size and shape of PVS and various levels of anatomic details in surrounding brain tissues, we integrating the multi-scale feature extraction into the encoder network of the proposed M2SS-PVS. As shown in Figure 1B, the first level of feature extraction consists of a convolutional and a pooling operation, and the second level reverses the order of the operations. In this manner, these multi-scale features are then fed into the corresponding encoder level. We design four multi-scale features in descending order to be fed into the encoder.

**Implementations and experiments.** The proposed networks were implemented with Python using the PyTorch deep learning library. The models were trained on Tesla V100 GPU. We randomly split the labeled subjects into training, validation, and testing sets (8:1:1) for 5-fold validation. Each subject was divided into mini patches with the size of 32x32x32 voxels (stride=16), which yielded 1540 patches for each subject. In supervised learning part, the model was fed only with labeled T1 and T2 images, while in semi-supervised learning part, M2SS-PVS was trained using both labeled and unlabeled images simultaneously. To alleviate class imbalance, we only chose the patches that include PVS voxels for a warm-up training (epoch=20).

**Evaluation.** We compared the results of M2SS-PVS with the supervised methods using the same encoder-decoder architecture 1) with and 2) without multi-scale feature extraction, and 3) semi-supervised methods without multi-scale feature extraction.

## 3. Results and conclusions

The results are summarized in Table 1. Our proposed network achieved the highest DSC and the lowest AVD compared to the other three networks. Mono-scale semi-supervised model yielded the highest recall but the $2^{nd}$ lowest DSC with the largest variance, suggesting its sensitivity to PVS, but inaccurate and inconsistent segmentation across individual PVSs. On the other hand, M2SS-PVS segmented PVS with the highest accuracy, highest consistency (lowest variance), and high sensitivity.

Table 1: The performance of networks in Dice similarity coefficient (DSC), absolute percentage volume difference (AVD), and Recall (mean ± standard deviation).

|  | DSC | AVD(%) | Recall |
|---|---|---|---|
| Supervised (mono-scale) | 0.50±0.15 | 55±11 | 0.70±0.19 |
| Supervised (multi-scale) | 0.56±0.44 | 51±83 | 0.54±0.44 |
| Semi-supervised (mono-scale) | 0.51±0.75 | 43±67 | **0.79±0.16** |
| M2SS-PVS (multi-scale) | **0.65±0.12** | **34±95** | 0.75±0.07 |

Our results demonstrate that the proposed method can improve PVS segmentation. We did not compare our network with other semi-supervised learning methods, yet, it is worth noting that our study was the first to employ a semi-supervised learning framework with modification to address the challenges involved in PVS segmentation task.

## Acknowledgments

This work is supported by the Alzheimer Disease Research Center (ADRC) at University of Southern California. Data were provided by the Human Connectome Project, WU-Minn Consortium (Principal Investigators: David Van Essen and Kamil Ugurbil; 1U54MH091657) funded by the 16 NIH Institutes and Centers that support the NIH Blueprint for Neuroscience Research; and by the McDonnell Center for Systems Neuroscience at Washington University.

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
