# OpenReview forum: "Semi-supervised learning in perivascular space segmentation using MRI images"
_MIDL.io/2023/Short_Paper_Track — MIDL 2023 Short paper track Poster_

### Official Review · Reviewer_FiZ8 · 2023-04-22
**A combination of supervised and semi-supervised learning for a difficult medical imaging task**

**Rating:** 6
**Confidence:** 3

**Review:**

The paper combines ideas from contrastive learning with supervised learning for segmentation.

Since this is a rare task and time-consuming to annotate, I would encourage the author to release their dataset openly.


Pros:
- PVS segmentation is a relatively rare problem in the literature

Cons:
- The writing and description of the encoder-decoder's training should be improved. For instance it is not obvious where the pseudo labels come from, or how the consistency loss and focal loss are combined.
- The supervised, multi-scale results shows very large variance in the Dice, which is not explained and somewhat surprising. This is typically indicating completely missed slices with very low or zero Dice, which can then bias results importantly. Is this the case?

---

### Official Review · Reviewer_jVA5 · 2023-04-26
**An interesting SSL approach for perivascular space segmentation**

**Rating:** 7
**Confidence:** 4

**Review:**

This paper investigates a 3D multi-channel (T1, T2 MRI), multi-scale semi-supervised perivascular space segmentation model. The model follows an encoder-decoder structure, with a focal loss for the supervised learning term. To leverage unlabeled samples with an unsupervised learning term, the authors deploy a loss evaluating the consistency between the pseudo label produced by T1 and T2 images and the predictions of  the augmented images. They further designed a multi-scale feature extraction in the encoder to incorporate different level of features in the successive layers.

The experiments show that both semi-supervision (i.e. using unlabeled samples) and multi-scale processing increase performances.

The paper is clear and well executed. It seems also that semi-supervision (i.e. leveraging unlabeled data) has not been explored before for this specific task. Therefore, I recommend acceptance.

Comment on the results in Table 1: Is there a reason why the supervised multi-scale approach is significantly lower in terms of recall in comparison to the supervised mono-scale approach. This is counterintuitive given that the multi-scale processing helps significantly on the semi-supervised setting.